# Longitudinal study of patients' health-related quality of life using EQ-5D-3L in 11 Swedish National Quality Registers

Fitsum Sebsibe Teni [![ORCID]] ,[1] Ola Rolfson,[1,2,3] Nancy Devlin,[4,5] David Parkin,[5,6] Emma Nauclér,[3] Kristina Burström,[1,7,8] The Swedish Quality Register (SWEQR) Study Group

For numbered affiliations see end of article.

**Correspondence to**
Fitsum Sebsibe Teni;
fitsum.teni@ki.se

## ABSTRACT

**Objective** To compare problems reported in the five EQ-5D-3L dimensions and EQ VAS scores at baseline and at 1-year follow-up among different patient groups and specific diagnoses in 11 National Quality Registers (NQRs) and to compare these with the general population.

**Design** Longitudinal, descriptive study.

**Participants** 2 66 241 patients from 11 NQRs and 49 169 participants from the general population were included in the study.

**Primary and secondary outcome measures** Proportions of problems reported in the five EQ-5D-3L dimensions, EQ VAS scores of participants' own health and proportions of participants and mean/median EQ VAS score in the Paretian Classification of Health Change (PCHC) categories.

**Results** In most of the included registers, and the general population, problems with pain/discomfort were the most frequently reported at baseline and at 1-year follow-up. Mean EQ VAS score (SD) ranged from 45.2 (22.4) among disc hernia patients to 88.1 (15.3) in wrist and hand fracture patients at baseline. They ranged from 48.9 (20.9) in pulmonary fibrosis patients to 83.3 (17.4) in wrist and hand fracture patients at follow-up. The *improved* category of PCHC, improvement in at least one dimension without deterioration in any other, accounted for the highest proportion in several diagnoses, corresponding with highest improvement in mean EQ VAS score.

**Conclusions** The study documented self-reported health of several different patient groups using the EQ-5D-3L in comparing with the general population. This demonstrated the important role of patient-reported outcomes in routine clinical care, to assess and follow-up health status and progress within different groups of patients. The EQ-5D-3L descriptive system and EQ VAS have an important role in providing a 'common denominator', allowing comparisons across NQRs and specific diagnoses.

**Trial registration number** ClinicalTrials.gov (NCT04359628).

## INTRODUCTION

Patient-reported outcomes (PROs) are increasingly used in healthcare.[1] The health-related quality of life (HRQoL) instrument EQ-5D[2] is a generic (not disease-specific) PRO measure used worldwide for a wide range of conditions and treatments. The EQ-5D

### Strengths and limitations of this study

► The study presents findings on a large number of diagnoses across many patient groups.
► Data on health-related quality of life from large National Quality Registers and from the general population were included.
► Change in health-related quality of life from baseline to 1-year follow-up was reported using a longitudinal design.
► Differences in health-related quality of life due to different modes of data collection are possible.
► The main analysis was restricted to patients with data both at baseline and 1-year follow-up.

descriptive system consists of five dimensions of health: mobility, self-care, usual activities, pain/discomfort and anxiety/depression.[3] The EQ VAS, a vertical line scaled from 0 ('worst imaginable health') to 100 ('best imaginable health') is a component of the EQ-5D.

Although EQ VAS plays a distinctive role in capturing patients' overall assessment of their own health, it is seldom the focus of analyses. A study in the UK that assessed routine collection of EQ VAS found a consistent relationship between problems reported on the EQ-5D dimensions and the EQ VAS score.[4] Another study found it feasible to use the EQ VAS in clinical routine in patient diaries.[5] EQ VAS has also been employed in studies as part of the EQ-5D instrument in the assessment of HRQoL among patients with different diagnoses and in different settings.[6–12]

The perception of problems experienced in the different dimensions and their responsiveness to healthcare intervention may differ depending on the nature and burden of the condition. Studies addressing these areas could provide useful information to clinicians as well as policymakers in guiding the use of the EQ-5D descriptive system,

including the EQ VAS component in patient care which, in some circumstances, has led to improved care and patient outcome.[13] Improved care could be achieved through increased understanding of patients' conditions and monitoring improvements after treatment, with due consideration for PROs.

In Sweden, National Quality Registers (NQRs) hold data on individual patients' diseases treatment/intervention and outcomes.[14] There are over 100 NQRs across a wide range of clinical areas, dating as far back as 1975, when the Swedish Knee Arthroplasty Register was established. Most of the NQRs have been initiated by healthcare professionals.[15] A number of clinical areas are covered by the NQRs, with the aim of helping provide patients with the best possible care through, for example, monitoring of healthcare performance and quality, ongoing learning and research.[16] The use of a personal identification number in Sweden provides an advantage when linking data on patients from the NQRs with those collected within other government-administered registers, such as the National Patient Register.[14] There have been a number of efforts to use the various NQRs for quality improvement.[17] Most of the NQRs collect PRO data, the most common generic HRQoL instrument being the EQ-5D, which is included in 40 registers.[15] Generic measures provide a means of comparing health problems among patients in different NQRs, to understand the overall severity of problems experienced by patients with different diseases and treatment pathways. Many studies have included EQ VAS as part of the assessment of HRQoL using the EQ-5D among patients in NQRs. Examples of such studies include the influence of different surgical approaches to the hip on HRQoL,[18] comparison after out-of-hospital and in-hospital cardiopulmonary resuscitation[19] as well as after hip replacement among patients born in Sweden and abroad[20] and self-reported health in patients with congenital heart diseases.[21]

The present descriptive study provides comprehensive information on HRQoL, using the three-level EQ-5D-3L (no, some/moderate, confined to bed/unable to/extreme problems), resulting in 243 health profiles or health states, and the EQ VAS, of several patient groups with various diagnoses, a subject that has not been extensively examined in previous research. The overall aim of this study was to compare problems reported in the five EQ-5D-3L dimensions and EQ VAS scores at baseline and 1-year follow-up among different patient groups and specific diagnoses in 11 NQRs and to compare these with the general population. The objectives were:

► To document the prevalence of problems reported according to level of severity within the EQ-5D-3L dimensions and EQ VAS scores among different patient groups and diagnoses.
► To document changes in the prevalence of problems reported according to level of severity within the EQ-5D-3L dimensions, and the change in EQ VAS scores among different patient groups and diagnoses.

► To relate the prevalence of problems and mean EQ VAS scores to the characteristics of patients, diagnoses and interventions, at baseline, at 1-year follow-up, and changes over the follow-up and to make comparisons with the general population.
► To calculate time-trade-off (TTO) and VAS indices for the diagnoses and to establish EQ-5D-3L reference data for patients with the different diagnoses.

## METHODS
### Study design
This is a descriptive study of longitudinal cohorts from 11 Swedish NQRs and comparison with data from the general population. This study forms part of a research project on the use of the EQ-5D questionnaire in different patient groups in Sweden.[22] The study is reported in line with the REporting of studies Conducted using Observational Routinely collected health Data (RECORD) checklist.[23]

### Swedish NQRs included in the study
The NQRs included in this study comprise a selection of different disease areas where EQ-5D-3L data are collected. In addition to musculoskeletal conditions, where EQ-5D-3L is used mostly, our sample also represents patients with heart failure, respiratory failure and psoriasis. The Better management of patients with Osteoarthritis (BOA) register is referred to as BOA throughout the manuscript (online supplemental table S1).

Data from the registers on PROs, body mass index (BMI) and demographic characteristics, at baseline, and at 1-year follow-up, were included. A 1-year follow-up was chosen as it is the most common evaluation time in the NQRs, which facilitated comparison between them. In registers with no such information, we used the date of data collection to determine the time points. EQ-5D-3L data collected within 90 days of one calendar year after baseline were categorised as 1-year follow-up data, following the approach employed in the Swedish Hip Arthroplasty Register. In the case of patients from the fracture register, baseline data on EQ-5D-3L were collected within 4 weeks from the fracture date, by recall of the week before the fracture occurred.

### Sampling and data
Baseline/first visit and 1-year follow-up data from 2001 to 2019 were retrieved from the registers. Patient records including EQ-5D-3L questionnaire data were selected. Records with complete data on main diagnosis/diagnosis groups and interventions were selected among these. Prior to pooling data from the registers, patients with records in more than one register were randomly deduplicated. After the exclusion of duplicate records, a total of 266 241 patient records, with complete data on age and sex at baseline, on diagnosis and intervention, and on EQ-5D-3L dimensions were included in the analysis of EQ-5D-3L dimensions. In the analysis of EQ VAS data, 209 247 records with complete data were included (online supplemental table S1).

The patient data were compared with the general population, using data on 49 169 individuals with complete data on EQ-5D-3L dimensions and 41 761 with complete EQ VAS data. This data were employed to develop the Swedish experience-based EQ-5D-3L value sets.[24] The general population data came from population surveys conducted in two regions in Sweden, Scania in 2004 and Stockholm in 2006, they are in terms of basic characteristics broadly representative of the Swedish population.[24] The self-administered questionnaires covered around 100 questions on living conditions and self-reported health of participants.[24–27] In the present study, for comparison with the findings from the different patient groups, data on demographic and the EQ-5D-3L questionnaire on 49 169 participants were included.

Age was categorised into <30, 30–39, 40–49, 50–59, 60–69, 70–79 and ≥80 years. BMI (weight in kilograms divided by a square metres of height) was categorised as underweight (<18.5), normal weight (18.5–24.9), overweight (25.0–29.9) and obesity class I (30–34.9), II (35.0–39.9) or III (40+).[28]

### Data analysis

Descriptive analysis was performed on age, sex, BMI categories and data on EQ-5D-3L dimensions. The prevalence of problems reported by patients was presented by register (patient group) and by diagnosis, at baseline and 1-year follow-up, and for data from the general population. In addition, to make comparisons, taking age and sex difference into account, problems reported among patients in the different registers and the general population were presented for women and men categorised by age groups.

Descriptive statistics (mean (SD) and median (IQR)) were also used to present EQ VAS score at baseline, at 1-year follow-up as well as for the change in score, by register, diagnosis, intervention and by age group, sex and BMI category in each register. An analysis of covariance (ANCOVA) was performed to make comparisons of the estimated mean EQ VAS scores by sex, adjusted for age in each register.

For the data at baseline and at 1-year follow-up, a two-level random intercept model (as there could be variations based on the register a diagnosis is grouped in) was conducted to assess how each diagnosis in the different registers influences mean EQ VAS score with the general population as a reference group. This analysis was performed on the pooled (combined) data of all the patient groups and the general population for each of the baseline and the 1-year follow-up. The estimates were adjusted for age group and sex. In the statistical analyses, a 95% CI was employed.

The methods described above for analyses of the NQR data were applied on the general population data.

Paretian Classification of Health Change (PCHC), which categorises changes in health states over time into *no problem*, *no change*, *improved*, *worsened* and *mixed*, was performed using baseline and 1-year follow-up data on health states.[29] Individuals who reported the same health state at both time points (including, but not limited to, health state 11111, ie, no problems in any of the dimensions) were included in the *no change* category. Improvement in any of the EQ-5D-3L dimensions with no deterioration in the remaining ones leads to categorisation as *improved*. On the contrary, a worsening in any of the EQ-5D-3L dimensions with no improvement in the remaining dimensions was categorised as *worsened*. Health states characterised by mixed types of changes (both improved and worsened) were grouped in the *mixed* category.[29] By using this categorisation, the proportion of improvements and worsening, as well as the other categories, was calculated for each diagnosis. Similarly, the mean self-assessed EQ VAS scores in these PCHC groups at baseline and 1-year follow-up were calculated according to diagnosis.

As a sensitivity analysis, the demographic characteristics of patients with complete baseline and 1-year follow-up data included in the analysis, and those with missing records at 1-year follow-up were compared for each register.

In order to assess the HRQoL profile of patients with complete data at baseline (regardless of follow-up status) and also patients with complete data at 1-year follow-up (regardless of their baseline status), prevalence of problems reported on the EQ-5D-3L dimensions and mean EQ VAS score was calculated. In addition, the EQ-5D-3L data were summarised as EQ-5D-3L indices using the Swedish experience-based EQ-5D-3L TTO and VAS value sets.[24]

### Patient and public involvement

Neither patients nor the general public were involved in the design, conduct, reporting or dissemination plans of our research.

## RESULTS

### Demographic characteristics

A total of 266 241 patient records and 49 169 records from general population were included in the study (online supplemental table S1). The mean age of included participants ranged from 29.8 years in patients with cruciate ligament injury to 73.7 years for patients with respiratory failure and 72.8 years in heart failure patients (table 1). Most of the patients were in their 50s to 70s. However, among heart failure patients, nearly one-third were 80 years or older, and among the cruciate ligament injury patients, the majority were below 30 years of age. The mean age of the general population was 46.2 years (table 1). A majority of the participants were women, with the exception of patients with psoriasis, heart failure as well as ankle and cruciate ligament injury. In almost all of the NQRs, a majority of patients had a BMI in the normal weight and overweight categories (table 1; online supplemental table S2).

**Table 1** Demographic characteristics and BMI in the 11 NQRs and in the general population

| Variable | Intervention-based registers | | | | | | | Diagnosis-based registers | | | | General population |
|---|---|---|---|---|---|---|---|---|---|---|---|---|
| | Spine | Hip | Knee | Ankle | Cruciate ligament | BOA | Fracture | Heart failure | Respiratory failure | Psoriasis | Rheumatology | |
| | n=48 960 | n=90 660 | n=16 324 | n=700 | n=8430 | n=13 647 | n=50 892 | n=1436 | n=1050 | n=2680 | n=31 462 | n=49 169 |
| | % | % | % | % | % | % | % | % | % | % | % | % |
| Age in years (mean (SD)) | 58.8 (15.3) | 68.5 (10.3) | 68.9 (8.7) | 63.2 (11.7) | 29.8 (10.0) | 65.5 (9.1) | 62.1 (18.1) | 72.8 (11.1) | 73.7 (7.7) | 50.3 (14.8) | 54.3 (15.7) | 46.2 (15.3) |
| Age group | | | | | | | | | | | | |
| <30 | 1674 (3.4) | 122 (0.1) | 1 (0.0) | 10 (1.4) | 4964 (58.9) | 6 (0.0) | 3793 (7.5) | 2 (0.1) | – | 251 (9.4) | 2593 (8.2) | 8160 (16.6) |
| 30–39 | 4652 (9.5) | 625 (0.7) | 10 (0.1) | 21 (3.0) | 1797 (21.3) | 76 (0.6) | 2870 (5.6) | 7 (0.5) | 1 (0.1) | 422 (15.7) | 3483 (11.1) | 9708 (19.7) |
| 40–49 | 7695 (15.7) | 3394 (3.7) | 278 (1.7) | 52 (7.4) | 1299 (15.4) | 624 (4.6) | 4651 (9.1) | 37 (2.6) | 1 (0.1) | 581 (21.7) | 5353 (17.0) | 9964 (20.3) |
| 50–59 | 8964 (18.3) | 11 766 (13.0) | 2136 (13.1) | 145 (20.7) | 347 (4.1) | 2537 (18.6) | 8216 (16.1) | 137 (9.5) | 41 (3.9) | 616 (23.0) | 6721 (21.4) | 10 491 (21.3) |
| 60–69 | 11 835 (24.2) | 30 285 (33.4) | 5814 (35.6) | 233 (33.3) | 22 (0.3) | 5849 (42.9) | 11 918 (23.4) | 294 (20.5) | 265 (25.2) | 568 (21.2) | 6893 (25.1) | 7602 (15.5) |
| 70–79 | 10 999 (22.5) | 31 869 (35.2) | 6296 (38.6) | 215 (30.7) | 1 (0.0) | 3836 (28.1) | 10 799 (21.2) | 512 (35.7) | 494 (47.0) | 201 (7.5) | 4448 (14.1) | 2978 (6.1) |
| 80+ | 3141 (6.4) | 12 599 (13.9) | 1789 (11.0) | 24 (3.4) | – | 719 (5.3) | 8645 (17.0) | 447 (31.1) | 248 (23.6) | 41 (1.5) | 971 (3.1) | 266 (0.5) |
| Sex | | | | | | | | | | | | |
| Women | 25 328 (51.7) | 51 263 (56.5) | 9392 (57.5) | 311 (44.4) | 3772 (44.7) | 9844 (72.1) | 33 781 (66.4) | 507 (35.3) | 623 (59.3) | 994 (37.1) | 20 668 (65.7) | 27 700 (56.3) |
| BMI* | n=45 343 | n=69 206 | n=15 937 | n=447 | n=4331 | n=13 366 | – | – | n=868 | n=2620 | – | n=22 094 |
| Underweight | 264 (0.6) | 501 (0.7) | 26 (0.2) | 3 (0.7) | 14 (0.3) | 40 (0.3) | – | – | 118 (13.6) | 36 (1.4) | – | 461 (2.1) |
| Normal weight | 15 371 (33.9) | 22 103 (31.9) | 3106 (19.5) | 97 (21.7) | 2547 (58.8) | 3728 (27.9) | – | – | 361 (41.6) | 760 (29.0) | – | 12 599 (57.0) |
| Overweight | 19 823 (43.7) | 30 164 (43.6) | 7240 (45.4) | 200 (44.7) | 1466 (33.8) | 5875 (44.0) | – | – | 220 (25.3) | 999 (38.1) | – | 6837 (30.9) |
| Obesity class I | 8089 (17.8) | 12 596 (18.2) | 4217 (26.5) | 107 (23.9) | 252 (5.8) | 2672 (20.0) | – | – | 110 (12.7) | 534 (20.4) | – | 1683 (7.6) |
| Obesity class II | 1549 (3.4) | 3200 (4.6) | 1138 (7.1) | 32 (7.2) | 39 (0.9) | 801 (6.0) | – | – | 46 (5.3) | 202 (7.7) | – | 366 (1.7) |
| Obesity class III | 247 (0.5) | 642 (0.9) | 210 (1.3) | 8 (1.8) | 13 (0.3) | 250 (1.9) | – | – | 13 (1.5) | 89 (3.4) | – | 148 (0.7) |
| Missing (count) | 3617 | 21 454 | 387 | 253 | 4099 | 281 | – | – | 182 | 60 | – | 27 075 |

BMI categories: underweight:<18.5; normal weight: 18.5–24.9; overweight: 25.0–29.9; obesity class I: 30.0–34.9; obesity class II: 35.0–39.9; obesity class III:≥40.

*Percentage based on non-missing values.

BMI, body mass index; BOA, Better management of patients with Osteoarthritis ; NQRs, National Quality Registers.

## Problems reported on the EQ-5D-3L dimensions

Table 2 presents the prevalence of problems reported in the EQ-5D-3L dimensions across NQRs, at baseline and 1-year follow-up. At both time points, similar to the general population, most problems reported were in the pain/discomfort dimension, with the exception of the respiratory failure patients, where most problems reported concerned mobility. Patients in BOA and those with spine, ankle, knee and hip conditions had the highest prevalence of pain/discomfort of the NQRs included at baseline. At the 1-year follow-up, the highest proportion of problems reported was among BOA, rheumatology and spine patients. In most registers, at both baseline and 1-year follow-up, problems with mobility were the second most prevalent.

At both time points, the highest proportion of severe problems reported was regarding pain/discomfort. This was the case for all registers as well as in the general population, with the exception of patients with fracture and respiratory failure, for which the highest proportion of severe problems concerned the usual activity dimension. In addition, patients with cruciate ligament injury at 1-year follow-up reported the highest proportion of severe problems in the anxiety/depression dimension (table 2).

The prevalence of problems reported in the EQ-5D-3L dimensions by patients according to specific diagnoses across NQRs is presented at baseline and at 1-year follow-up (online supplemental table S3). For all diagnoses, with exception of chronic obstructive pulmonary disease (COPD) and pulmonary fibrosis, most problems were reported with pain/discomfort. This was most commonly seen at baseline in the intervention-based registers covering musculoskeletal diseases, with the highest proportion among patients concerning a diagnosis of rheumatoid arthritis in the knee (100%). Patients with rheumatoid arthritis of the ankle had the highest prevalence of severe problems in the pain/discomfort (69%) dimension. Among patients with COPD and pulmonary fibrosis, the highest prevalence of problems was in the mobility dimension. At 1-year follow-up, the highest proportion of severe problems was reported in the pain/discomfort dimension for most diagnoses, except for patients with lower extremity and axial fractures and for patients with COPD and pulmonary fibrosis.

Severe problems in the anxiety/depression dimension were most prevalent among patients with COPD, both at baseline (9%) and 1-year follow-up (10%). The response, 'confined to bed', within the mobility dimension was very uncommon among all diagnoses, with the largest proportion occurring in patients with disc hernia (5%) at baseline and those with fracture on hip and thigh (5%) as well as COPD and pulmonary fibrosis (5%) at 1-year follow-up. The prevalence of severe problems with self-care was highest at 1-year follow-up in patients with a hip or femoral fracture (10%) (online supplemental table S3).

Furthermore, the proportion of problems reported on the EQ-5D-3L among women and men categorised by age groups in the 11 NQRs at baseline and 1-year follow-up and in the general population is presented in online supplemental figures 1 and 2.

## Paretian Classification of Health Change

Figure 1 presents the proportions of patients by PCHC group for the different diagnoses. The highest proportions of improvements were shown for patients in the intervention-based ankle, hip, knee and spine registers. For most of the other diagnoses, the highest proportions of patients were classified as *improved*. The respiratory failure patient group was an exception, of whom most were classified as *worsened*. Another exception was the fracture patient group where the highest proportion of patients was classified as *worsened*, as their baseline data were collected by recall of the week before the fracture occurred. Most patients with fracture had a better health state at baseline than at 1-year follow-up. The heart failure patients had similar proportions of patients classified as *improved* and *worsened*.

## Mean self-assessed EQ VAS score

Mean (table 3) and median (online supplemental table S4) self-assessed EQ VAS scores by sex, age groups and BMI categories are described for each register at baseline, 1-year follow-up, the change over the period and for the general population. The highest mean score at both time points was found among fracture patients (84.7 (baseline) and 77.7 (1 year), respectively). The lowest scores were found among spine patients (47.8) at baseline and patients with respiratory failure (49.1) at 1-year follow-up. The mean EQ VAS score in the general population was 79.5. The scores were found to have increased in all patients at the 1-year follow-up, with the exception of respiratory failure patients and fracture patients. The highest increases were recorded among hip (20.5) and spine surgery (20.1) patients.

Similar to the general population, in most registers, women had statistically significantly lower mean EQ VAS scores after adjustment for age in the ANCOVA. Exceptions were seen among patients with fracture and respiratory failure at baseline and BOA, fracture, heart failure and respiratory failure patients at 1-year follow-up (table 4). The mean EQ VAS score showed greater improvements among women in most of the registers, with the exception of patients with ankle, cruciate ligament injury, heart and respiratory failure.

In most registers with BMI data, the highest mean EQ VAS scores were found in the normal and overweight categories, at both time points. However, no clear pattern was shown in terms of the change in mean EQ VAS score (table 3).

Figure 2 describes mean EQ VAS score by diagnosis at baseline and 1-year follow-up. Mean EQ VAS scores had more similarity among diagnoses within the same patient group (register) than with diagnoses in other patient

**Table 2** Prevalence of problems reported on EQ-5D-3L, at baseline and 1 year follow-up in the 11 NQRs and in the general population

| Patient group in NQRs/ | Severity level | EQ-5D-3L dimension | | | | | | | | | |
|---|---|---|---|---|---|---|---|---|---|---|---|
| | | Baseline | | | | | 1 year follow-up | | | | |
| | | Mobility % | Self-care % | Usual activities % | Pain/ discomfort % | Anxiety/ depression % | Mobility % | Self-care % | Usual activities % | Pain/ discomfort % | Anxiety/ depression % |
| **Intervention-based registers** | | | | | | | | | | | |
| Spine (n=48960) | Level 1 | 7397 (15.1) | 39123 (**79.9**) | 13691 (28.0) | 302 (0.6) | 20660 (42.2) | 27881 (**56.9**) | 44819 (**91.5**) | 32140 (**65.6**) | 11179 (22.8) | 30539 (**62.4**) |
| | Level 2 | 40622 (**83.0**) | 9251 (18.9) | 26059 (**53.2**) | 24076 (49.2) | 25351 (**51.8**) | 20950 (42.8) | 3857 (7.9) | 14637 (29.9) | 30954 (**63.2**) | 16377 (33.4) |
| | Level 3 | 941 (1.9) | 586 (1.2) | 9210 (18.8) | 24582 (**50.2**) | 2949 (6.0) | 129 (0.3) | 284 (0.6) | 2183 (4.5) | 6827 (13.9) | 2044 (4.2) |
| Hip (n=90660) | Level 1 | 7340 (8.1) | 70483 (**77.7**) | 36364 (40.1) | 1400 (1.5) | 54120 (**59.7**) | 55970 (**61.7**) | 83800 (**92.4**) | 70683 (**78.0**) | 41270 (45.5) | 71102 (**78.4**) |
| | Level 2 | 83026 (**91.6**) | 19307 (21.3) | 45371 (**50.0**) | 52519 (**57.9**) | 33597 (37.1) | 34565 (38.1) | 6319 (7.0) | 18161 (20.0) | 45422 (**50.1**) | 18188 (20.1) |
| | Level 3 | 294 (0.3) | 870 (1.0) | 8925 (9.8) | 36741 (40.5) | 2943 (3.2) | 125 (0.1) | 541 (0.6) | 1816 (2.0) | 3968 (4.4) | 1370 (1.5) |
| Knee (n=16324) | Level 1 | 1895 (11.6) | 15241 (**93.4**) | 8719 (**53.4**) | 290 (1.8) | 10669 (**65.4**) | 10249 (**62.8**) | 15522 (**95.1**) | 12696 (**77.8**) | 5884 (36.0) | 12830 (**78.6**) |
| | Level 2 | 14388 (**88.1**) | 938 (5.7) | 6742 (41.3) | 10328 (**63.3**) | 5261 (32.2) | 6048 (37.0) | 710 (4.3) | 3354 (20.5) | 9588 (**58.7**) | 3222 (19.7) |
| | Level 3 | 41 (0.3) | 145 (0.9) | 863 (5.3) | 5706 (35.0) | 394 (2.4) | 27 (0.2) | 92 (0.6) | 274 (1.7) | 852 (5.2) | 272 (1.7) |
| Ankle (n=700) | Level 1 | 29 (4.1) | 617 (**88.1**) | 264 (37.7) | 8 (1.1) | 404 (**57.7**) | 253 (36.1) | 641 (**91.6**) | 448 (**64.0**) | 170 (24.3) | 503 (**71.9**) |
| | Level 2 | 659 (**94.1**) | 76 (10.9) | 346 (**49.4**) | 351 (**50.1**) | 266 (38.0) | 443 (**63.3**) | 51 (7.3) | 223 (31.9) | 459 (**65.6**) | 180 (25.7) |
| | Level 3 | 12 (1.7) | 7 (1.0) | 90 (12.9) | 341 (48.7) | 30 (4.3) | 4 (0.6) | 8 (1.1) | 29 (4.1) | 71 (10.1) | 17 (2.4) |
| Cruciate ligament (n=8430) | Level 1 | 5645 (**67.0**) | 8209 (**97.4**) | 4503 (**53.4**) | 1294 (15.3) | 4190 (**49.7**) | 7308 (**86.7**) | 8337 (**98.9**) | 6663 (**79.0**) | 2943 (34.9) | 5501 (**65.3**) |
| | Level 2 | 2762 (32.8) | 178 (2.1) | 3172 (37.6) | 6658 (**79.0**) | 3779 (44.8) | 1117 (13.3) | 69 (0.8) | 1636 (19.4) | 5223 (**62.0**) | 2649 (31.4) |
| | Level 3 | 23 (0.3) | 43 (0.5) | 755 (9.0) | 478 (5.7) | 461 (5.5) | 5 (0.1) | 24 (0.3) | 131 (1.6) | 264 (3.1) | 280 (3.3) |
| BOA (n=13647) | Level 1 | 5781 (42.4) | 13120 (**96.1**) | 10274 (**75.3**) | 351 (2.6) | 8981 (**65.8**) | 7096 (**52.0**) | 13093 (**95.9**) | 10880 (**79.7**) | 1264 (9.3) | 9553 (**70.0**) |
| | Level 2 | 7854 (**57.6**) | 470 (3.4) | 3200 (23.4) | 11755 (**86.1**) | 4473 (32.8) | 6542 (47.9) | 494 (3.6) | 2646 (19.4) | 11278 (**82.6**) | 3930 (28.8) |
| | Level 3 | 12 (0.1) | 57 (0.4) | 173 (1.3) | 1541 (11.3) | 193 (1.4) | 9 (0.1) | 60 (0.4) | 121 (0.9) | 1105 (8.1) | 164 (1.2) |
| Fracture (n=50892) | Level 1 | 41022 (**80.6**) | 45512 (**89.4**) | 42073 (**82.7**) | 32363 (**63.6**) | 40462 (**79.5**) | 35314 (**69.4**) | 44852 (**88.1**) | 38161 (**75.0**) | 20558 (40.4) | 38181 (**75.0**) |
| | Level 2 | 9542 (18.7) | 4373 (8.6) | 5815 (11.4) | 16658 (32.7) | 9419 (18.5) | 14970 (29.4) | 4575 (9.0) | 9864 (19.4) | 28059 (**55.1**) | 11566 (22.7) |
| | Level 3 | 328 (0.6) | 1007 (2.0) | 3004 (5.9) | 1871 (3.7) | 1011 (2.0) | 608 (1.2) | 1465 (2.9) | 2867 (5.6) | 2275 (4.5) | 1145 (2.2) |
| **Diagnosis-based registers** | | | | | | | | | | | |
| Heart failure (n=1436) | Level 1 | 866 (60.3) | 1315 (**91.6**) | 1041 (**72.5**) | 689 (**48.0**) | 865 (**60.2**) | 837 (**58.3**) | 1298 (**90.4**) | 1025 (**71.4**) | 685 (**47.7**) | 896 (**62.4**) |
| | Level 2 | 567 (39.5) | 116 (8.1) | 355 (24.7) | 679 (47.3) | 521 (36.3) | 593 (41.3) | 126 (8.8) | 371 (25.8) | 655 (45.6) | 499 (34.7) |
| | Level 3 | 3 (0.2) | 5 (0.3) | 40 (2.8) | 68 (4.7) | 50 (3.5) | 6 (0.4) | 12 (0.8) | 40 (2.8) | 96 (6.7) | 41 (2.9) |

Continued

**Table 2** Continued

| Patient group in NQRs/ | Severity level | Baseline | | | | | 1 year follow-up | | | | |
|---|---|---|---|---|---|---|---|---|---|---|---|
| | | Mobility | Self-care | Usual activities | Pain/ discomfort | Anxiety/ depression | Mobility | Self-care | Usual activities | Pain/ discomfort | Anxiety/ depression |
| | | % | % | % | % | % | % | % | % | % | % |
| Respiratory failure (n=1050) | Level 1 | 243 (23.1) | 650 (**61.9**) | 272 (25.9) | 305 (29.0) | 432 (41.1) | 187 (17.8) | 551 (**52.5**) | 211 (20.1) | 256 (24.4) | 387 (36.9) |
| | Level 2 | 786 (**74.9**) | 343 (32.7) | 557 (**53.0**) | 600 (**57.1**) | 532 (**50.7**) | 814 (**77.5**) | 409 (39.0) | 526 (**50.1**) | 627 (**59.7**) | 559 (**53.2**) |
| | Level 3 | 21 (2.0) | 57 (5.4) | 221 (21.0) | 145 (13.8) | 86 (8.2) | 49 (4.7) | 90 (8.6) | 313 (29.8) | 167 (15.9) | 104 (9.9) |
| Psoriasis (n=2680) | Level 1 | 2064 (**77.0**) | 2554 (**95.3**) | 2183 (**81.5**) | 885 (33.0) | 1318 (**49.2**) | 2151 (**80.3**) | 2578 (**96.2**) | 2329 (86.**9**) | 1231 (45.9) | 1725 (**64.4**) |
| | Level 2 | 615 (22.9) | 115 (4.3) | 454 (16.9) | 1536 (**57.3**) | 1185 (44.2) | 526 (19.6) | 95 (3.5) | 312 (11.6) | 1307 (**48.8**) | 847 (31.6) |
| | Level 3 | 1 (0.0) | 11 (0.4) | 43 (1.6) | 259 (9.7) | 177 (6.6) | 3 (0.1) | 7 (0.3) | 39 (1.5) | 142 (5.3) | 108 (4.0) |
| Rheumatology (n=31 462) | Level 1 | 15 883 (**50.5**) | 25 394 (**80.7**) | 18 304 (**58.2**) | 3891 (12.4) | 16 917 (**53.8**) | 17 585 (**55.9**) | 26 523 (**84.3**) | 20 443 (**65.0**) | 5373 (17.1) | 18 408 (**58.5**) |
| | Level 2 | 15 506 (49.3) | 5718 (18.2) | 11 645 (37.0) | 22 703 (**72.2**) | 13 014 (41.4) | 13 776 (**43.8**) | 4591 (14.6) | 9873 (31.4) | 22 206 (**70.6**) | 11 553 (36.7) |
| | Level 3 | 73 (0.2) | 350 (1.1) | 1513 (4.8) | 4868 (15.5) | 1531 (4.9) | 101 (0.3) | 348 (1.1) | 1146 (3.6) | 3883 (12.3) | 1501 (4.8) |
| General population (n=49 169) | Level 1 | 44 279 (**90.1**) | 48 371 (**98.4**) | 44 848 (**91.2**) | 24 946 (**50.8**) | 32 721 (**66.5**) | – | – | – | – | – |
| | Level 2 | 4840 (9.8) | 600 (1.2) | 3785 (7.7) | 22 185 (45.1) | 15 126 (30.8) | – | – | – | – | – |
| | Level 3 | 50 (0.1) | 198 (0.4) | 536 (1.1) | 2038 (4.1) | 1322 (2.7) | – | – | – | – | – |

The highest proportion in bold; level 1: no problems; level 2: some/moderate problems; level 3: confined to bed/unable to/extreme problems; proportions rounded off to one decimal using MS Excel.
EQ-5D-3L is the three-level severity version of the EQ-5D questionnaire.
BOA, Better management of patients with Osteoarthritis; NQRs, National Quality Registers.

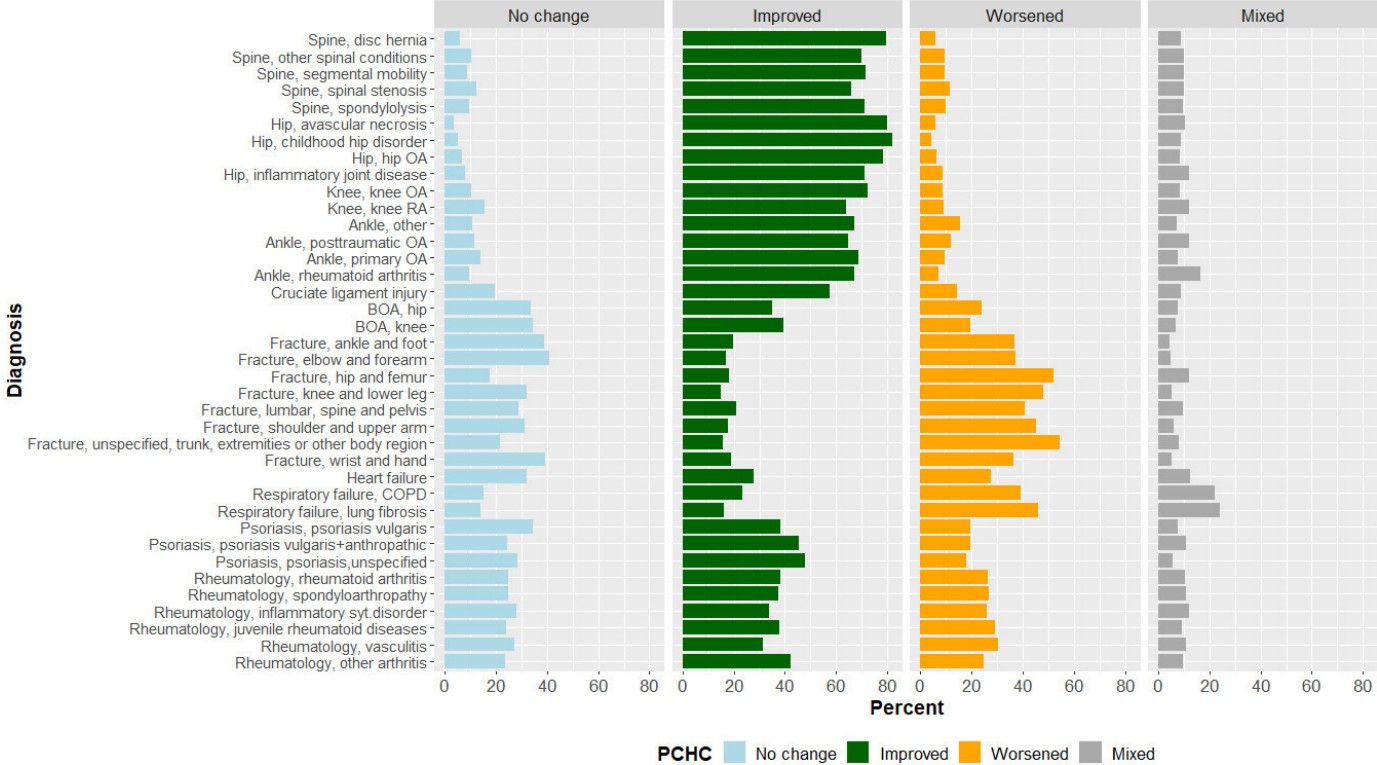

**Figure 1** Proportion of patients by Paretian Classification of Health Change (PCHC), by diagnosis in the 11 National Quality Registers (NQRs). BOA, Better management of patients with Osteoarthritis; COPD, chronic obstructive pulmonary disease; OA, osteoarthritis; RA, rheumatoid arthritis.

groups. Some differences were shown by diagnosis in some patient groups such as fracture (eg, hip and thigh) and hip (eg, inflammatory joint disorder), which were illustrated to be statistically significant in the results of the two-level random intercept model in figure 3. For most of the diagnoses, EQ VAS scores increased over the follow-up period; however, they decreased among patients with respiratory failure and fracture.

Figure 3 illustrates the two-level random intercept model estimates of mean EQ VAS scores for the different diagnoses, adjusted for age and sex, with the general population as a reference group. At baseline, patients with diagnoses other than fractures showed statistically significantly lower mean EQ VAS scores as compared with the general population. At 1-year follow-up, all patients, except those with BOA hip, fracture and respiratory failure diagnosis showed improvement in mean EQ VAS, while remaining lower than in the general population with the exception of fracture patients. A table form of the regression analysis is available in online supplemental table S5.

Figure 4 presents mean EQ VAS scores, at baseline and 1-year follow-up, in the various PCHC categories within the different diagnoses. Mean EQ VAS scores showed corresponding increases and decreases in the categories, improved and worsened. The largest increases in the improved category were found among diagnoses in the hip and spine patient groups. In the worsened category, diagnoses in the fracture patient groups showed the largest

decrements, from recall of their health status before the fracture occurred. In the categories, no change and mixed, increase in mean EQ VAS score was found in all diagnoses, with the exception of the fracture and respiratory failure patient groups, which instead showed decrements.

A comparison of patient records with complete data at baseline and at 1-year follow-up with data from patients with only baseline showed that higher proportions of women and older patients had complete baseline and 1-year follow-up data in many of the registers (results available on request).

Problems reported on EQ-5D-3L dimensions and mean self-assessed EQ VAS score among patients at baseline (regardless of follow-up status), among those with 1-year follow-up data (regardless of baseline status) and for the general population were calculated (online supplemental tables S6 and S7). The tables also present EQ-5D-3L indices based on the Swedish experience-based TTO and VAS value sets.[26] The results presented in Table S6 can serve as EQ-5D-3L reference data for patients with these diagnoses, with the exception of the fracture patients as their baseline data come from before the fracture date. The pattern of problems reported, and mean EQ VAS scores, were comparable to those reported at the corresponding time point among patients with data both at baseline and at 1-year follow-up.

**Table 3** Mean (SD) EQ VAS score at baseline, 1-year follow-up and the change in the nine NQRs and in the general population

| | | Mean EQ VAS score (SD) | | | | | | | | | |
|---|---|---|---|---|---|---|---|---|---|---|---|
| | | Intervention-based registers | | | | | | | Diagnosis-based registers | | |
| Time | Variable | Spine | Hip | Knee | Ankle | Cruciate ligament | BOA | Fracture | Heart failure | Respiratory failure | General population |
| Baseline | Overall | 47.8 (22.1) | 56.2 (22.2) | 64.6 (22.1) | 55.8 (21.4) | 62.8 (23.0) | 68.3 (18.6) | 84.7 (18.4) | 65.4 (18.9) | 51.1 (21.3) | 79.5 (18.3) |
| | Sex | | | | | | | | | | |
| | Men | 50.4 (22.1) | 59.6 (21.3) | 68.1 (20.9) | 58.5 (20.5) | 63.9 (23.0) | 70.8 (17.4) | 86.1 (17.0) | 66.5 (18.9) | 50.9 (20.2) | 80.4 (17.4) |
| | Women | 45.2 (21.8) | 53.7 (22.5) | 62.0 (22.5) | 52.5 (22.2) | 61.4 (23.0) | 67.3 (18.9) | 84.0 (19.0) | 62.9 (18.7) | 51.2 (22.0) | 78.8 (18.9) |
| | Age group | | | | | | | | | | |
| | <30 | 46.6 (21.3) | 46.2 (24.3) | – | 49.9 (17.5) | 63.3 (23.1) | 71.3 (2.5) | 90.3 (13.8) | – | – | 81.2 (16.8) |
| | 30–39 | 45.1 (22.0) | 52.2 (22.3) | 58.9 (25.6) | 45.6 (18.1) | 63.4 (22.4) | 62.6 (22.2) | 88.8 (14.7) | 74.0 (20.7) | – | 80.9 (17.1) |
| | 40–49 | 45.9 (21.7) | 54.3 (22.5) | 56.7 (23.3) | 51.4 (21.3) | 60.7 (23.5) | 63.9 (19.6) | 88.3 (15.6) | 67.2 (16.6) | – | 79.4 (18.3) |
| | 50–59 | 46.8 (22.3) | 55.0 (22.6) | 60.6 (23.4) | 55.0 (20.7) | 60.7 (22.8) | 66.1 (19.5) | 87.6 (16.3) | 62.1 (19.7) | 47.4 (27.8) | 78.5 (19.2) |
| | 60–69 | 48.7 (22.2) | 56.8 (22.6) | 64.5 (22.5) | 56.4 (21.8) | 59.0 (28.0) | 69.3 (18.3) | 87.4 (16.4) | 69.5 (19.8) | 50.0 (21.2) | 78.5 (19.4) |
| | 70–79 | 49.9 (22.2) | 57.1 (22.7) | 66.3 (21.3) | 58.7 (22.0) | – | 69.3 (18.0) | 83.3 (18.6) | 65.2 (19.4) | 50.7 (20.0) | 76.1 (19.3) |
| | 80+ | 49.6 (21.7) | 54.8 (21.4) | 64.8 (20.7) | 50.3 (17.6) | – | 66.6 (18.1) | 71.2 (22.1) | 63.2 (16.8) | 53.8 (22.5) | 71.1 (22.3) |
| | BMI | | | | | | | | | | |
| | Underweight | 42.6 (22.0) | 53.6 (22.1) | 62.3 (24.9) | 40.7 (29.9) | 72.6 (20.6) | 64.8 (20.4) | – | – | 48.9 (21.6) | 78.2 (19.1) |
| | Normal weight | 48.1 (22.2) | 57.9 (22.2) | 66.7 (21.9) | 56.1 (23.3) | 64.8 (23.0) | 71.1 (17.9) | – | – | 52.5 (21.9) | 82.1 (16.1) |
| | Overweight | 48.6 (22.0) | 57.6 (21.9) | 66.3 (21.7) | 57.4 (19.8) | 63.7 (22.5) | 69.2 (18.2) | – | – | 52.0 (19.1) | 78.7 (17.9) |
| | Obesity class I | 46.8 (21.7) | 54.3 (22.1) | 62.4 (21.9) | 54.5 (20.9) | 59.2 (23.3) | 65.1 (19.2) | – | – | 51.4 (21.2) | 72.9 (20.3) |
| | Obesity class II | 43.8 (22.3) | 51.6 (22.8) | 58.7 (23.1) | 53.0 (19.9) | 51.4 (24.2) | 63.2 (18.8) | – | – | 54.1 (19.2) | 68.4 (21.1) |
| | Obesity class III | 44.0 (21.8) | 49.4 (23.6) | 57.5 (23.5) | 40.5 (14.5) | 66.2 (20.2) | 56.7 (19.3) | – | – | 55.6 (17.3) | 67.5 (21.9) |

Continued

**Table 3** Continued

| Time | Variable | Mean EQ VAS score (SD) | | | | | | | | | |
| | | Intervention-based registers | | | | | | | Diagnosis-based registers | | |
| | | Spine | Hip | Knee | Ankle | Cruciate ligament | BOA | Fracture | Heart failure | Respiratory failure | General population |
| 1-year follow-up | Overall | 67.9 (22.5) | 76.7 (20.0) | 76.2 (19.4) | 70.2 (18.9) | 74.5 (19.9) | 70.4 (18.7) | 77.7 (20.4) | 67.9 (18.8) | 49.1 (20.4) | – |
| | Sex | | | | | | | | | | – |
| | Men | 69.4 (22.0) | 78.6 (19.0) | 77.8 (18.8) | 72.3 (17.1) | 75.3 (20.1) | 70.8 (18.3) | 78.7 (19.8) | 68.4 (19.0) | 49.4 (19.9) | – |
| | Women | 66.5 (22.9) | 75.3 (20.7) | 75.1 (19.8) | 67.6 (20.5) | 73.6 (19.5) | 70.3 (18.9) | 77.2 (20.8) | 65.4 (18.2) | 48.9 (20.7) | – |
| | Age group | | | | | | | | | | – |
| | <30 | 73.2 (20.3) | 74.2 (20.1) | – | 62.5 (23.7) | 74.5 (19.8) | 69.5 (28.2) | 82.5 (17.1) | – | – | – |
| | 30–39 | 70.9 (21.7) | 77.4 (20.4) | 75.5 (20.6) | 64.4 (20.4) | 74.5 (19.5) | 71.7 (20.9) | 82.0 (17.4) | 76.0 (19.5) | – | – |
| | 40–49 | 70.5 (22.6) | 78.8 (19.5) | 73.3 (20.7) | 67.3 (17.4) | 74.8 (20.4) | 69.8 (19.1) | 81.2 (17.7) | 71.7 (17.4) | – | – |
| | 50–59 | 68.3 (23.1) | 78.5 (19.8) | 73.6 (20.8) | 69.4 (20.2) | 73.4 (21.1) | 69.5 (19.7) | 80.7 (18.6) | 70.5 (18.1) | 48.8 (22.7) | – |
| | 60–69 | 68.5 (22.3) | 79.1 (19.4) | 78.0 (18.6) | 72.6 (17.2) | 79.7 (19.1) | 71.9 (18.1) | 81.3 (18.3) | 72.1 (17.8) | 47.7 (20.5) | – |
| | 70–79 | 64.8 (22.3) | 75.8 (20.1) | 76.5 (19.4) | 71.0 (18.6) | – | 69.9 (18.5) | 76.1 (21.0) | 66.6 (18.4) | 49.1 (19.6) | – |
| | 80+ | 60.7 (21.5) | 71.1 (20.5) | 73.1 (19.6) | 58.6 (23.0) | – | 64.9 (19.3) | 63.1 (23.0) | 63.0 (19.5) | 50.3 (21.4) | – |
| | BMI | | | | | | | | | | |
| | Underweight | 66.2 (24.0) | 72.7 (22.0) | 70.3 (16.9) | 51.7 (16.1) | 70.8 (19.8) | 69.4 (14.7) | – | – | 44.9 (20.7) | – |
| | Normal weight | 70.3 (22.2) | 78.6 (19.6) | 77.9 (18.9) | 68.9 (19.2) | 75.6 (20.0) | 73.9 (17.6) | – | – | 49.9 (20.5) | – |
| | Overweight | 68.6 (22.1) | 77.7 (19.4) | 77.6 (18.9) | 72.4 (17.6) | 74.6 (20.5) | 70.9 (18.4) | – | – | 50.9 (20.2) | – |
| | Obesity class I | 64.5 (22.9) | 74.5 (20.6) | 74.4 (19.7) | 67.3 (18.4) | 72.1 (20.9) | 66.9 (19.3) | – | – | 48.5 (20.8) | – |
| | Obesity class II | 60.3 (23.3) | 70.9 (21.6) | 72.7 (20.2) | 69.1 (22.8) | 64.8 (25.6) | 65.7 (18.9) | – | – | 47.2 (16.0) | – |
| | Obesity class III | 61.4 (24.8) | 68.1 (23.0) | 66.7 (22.6) | 39.4 (18.6) | 61.8 (27.2) | 58.6 (21.1) | – | – | 54.3 (15.1) | – |

Continued

**Table 3** Continued

| Time | Variable | Mean EQ VAS score (SD) | | | | | | | | | |
| | | Intervention-based registers | | | | | | Diagnosis-based registers | | | |
| | | Spine | Hip | Knee | Ankle | Cruciate ligament | BOA | Fracture | Heart failure | Respiratory failure | General population |
| Change | Overall | 20.1 (27.1) | 20.5 (25.8) | 11.7 (24.6) | 14.4 (24.7) | 11.8 (26.4) | 2.1 (18.7) | −7.0 (19.4) | 2.1 (18.3) | −2.0 (24.1) | – |
| | Sex | | | | | | | | | | – |
| | Men | 19.0 (26.7) | 19.0 (24.5) | 9.7 (23.5) | 13.8 (22.5) | 11.4 (26.4) | 0.0 (17.3) | −7.4 (18.9) | 1.9 (18.3) | −1.4 (24.1) | – |
| | Women | 21.2 (26.4) | 21.6 (26.7) | 13.1 (25.2) | 15.1 (27.2) | 12.2 (26.3) | 3.0 (19.2) | −6.8 (19.6) | 2.5 (18.2) | −2.3 (24.2) | – |
| | Age group | | | | | | | | | | – |
| | <30 | 26.6 (26.3) | 28.0 (29.0) | – | 12.6 (32.3) | 11.3 (26.6) | −1.8 (27.5) | −7.8 (19.5) | – | – | – |
| | 30–39 | 25.9 (27.3) | 25.2 (25.4) | 16.6 (21.3) | 18.8 (17.8) | 11.2 (24.6) | 9.0 (17.9) | −6.8 (18.3) | 2.0 (11.0) | – | – |
| | 40–49 | 24.6 (27.4) | 24.5 (26.0) | 16.6 (27.4) | 15.9 (25.7) | 14.1 (27.2) | 5.9 (19.0) | −7.1 (18.4) | 4.5 (19.2) | – | – |
| | 50–59 | 21.4 (27.2) | 23.5 (25.9) | 13.0 (26.0) | 14.4 (25.1) | 12.7 (27.9) | 3.4 (19.2) | −6.9 (18.8) | 8.4 (19.1) | 1.4 (26.1) | – |
| | 60–69 | 19.8 (26.7) | 22.3 (25.8) | 13.6 (24.6) | 16.2 (25.0) | 20.6 (31.4) | 2.6 (18.6) | −6.1 (18.2) | 2.6 (17.5) | −2.3 (22.9) | – |
| | 70–79 | 15.0 (26.1) | 18.7 (25.5) | 10.2 (23.9) | 12.3 (24.3) | – | 0.6 (18.4) | −7.2 (19.7) | 1.4 (19.2) | −1.6 (23.9) | – |
| | 80+ | 11.1 (25.8) | 16.3 (25.6) | 8.4 (23.8) | 8.3 (23.3) | – | −1.7 (17.8) | −8.1 (22.3) | −0.3 (16.8) | −3.5 (25.8) | – |
| | BMI | | | | | | | | | | – |
| | Underweight | 23.6 (29.0) | 19.1 (26.1) | 8.0 (24.2) | 11.0 (13.9) | −1.7 (22.0) | 4.6 (20.9) | – | – | −4.0 (23.5) | – |
| | Normal weight | 22.2 (27.4) | 20.8 (25.5) | 11.2 (23.5) | 12.8 (25.9) | 10.8 (26.6) | 2.8 (17.6) | – | – | −2.6 (24.2) | – |
| | Overweight | 20.0 (26.7) | 20.1 (25.3) | 11.3 (24.1) | 14.9 (23.1) | 10.9 (27.2) | 1.8 (18.5) | – | – | −1.1 (23.0) | – |
| | Obesity class I | 17.7 (27.2) | 20.2 (26.4) | 12.0 (25.2) | 12.8 (23.9) | 12.9 (27.1) | 1.8 (20.0) | – | – | −2.9 (24.6) | – |
| | Obesity class II | 16.5 (26.9) | 19.3 (27.4) | 14.0 (26.7) | 16.1 (34.0) | 13.4 (28.5) | 2.5 (20.3) | – | – | −6.9 (21.0) | – |
| | Obesity class III | 17.3 (28.7) | 18.7 (28.6) | 9.2 (28.5) | −1.1 (17.0) | −4.3 (21.1) | 1.9 (18.6) | – | – | −1.3 (13.6) | – |

BMI categories: underweight:<18.5; normal weight: 18.5–24.9; overweight: 25.0–29.9; obesity class I: 30.0–34.9; obesity class II: 35.0–39.9; obesity class III:≥40.

EQ VAS is the visual anaogue scale component of the EQ-5D questionnaire.

BMI, body mass index; BOA, Better management of patients with Osteoarthritis; NQRs, National Quality Registers.

**Table 4** Analysis of covariance testing difference in mean EQ VAS score by sex adjusted for age at baseline, 1 year and change in the nine NQRS and in the general population

| Time | Variable | Sex | Adjusted mean EQ VAS score (95% CI) | | | | | | | | | |
| | | | Intervention-based registers | | | | | | Diagnosis-based registers | | | |
| | | | Spine | Hip | Knee | Ankle | Cruciate ligament | BOA | Fracture | Heart failure | Respiratory failure | General population |
| Baseline | Sex | Men | 50.5 (50.2 to 50.8) | 59.7 (59.5 to 59.9) | 68.1 (67.6 to 68.6) | 58.4 (56.2 to 60.6) | 63.9 (63.2 to 64.6) | 70.7 (69.9 to 71.5) | 84.7 (84.4 to 85.0) | 66.5 (65.1 to 67.9) | 50.8 (48.3 to 53.3) | 80.6 (80.3 to 80.8) |
| | | Women | 45.2 (44.9 to 45.5) | 53.6 (53.4 to 53.8) | 62.0 (61.5 to 62.4) | 52.6 (50.2 to 55.0) | 61.4 (60.6 to 62.1) | 67.3 (66.8 to 67.8) | 84.7 (84.5 to 84.9) | 63.0 (60.9 to 65.0) | 51.2 (49.2 to 53.2) | 78.7 (78.4 to 78.9) |
| 1-year follow-up | Sex | Men | 69.3 (69.0 to 69.6) | 78.3 (78.1 to 78.5) | 77.8 (77.4 to 78.3) | 72.3 (70.4 to 74.2) | 75.3 (74.8 to 75.9) | 70.9 (70.1 to 71.7) | 77.2 (76.9 to 77.5) | 68.2 (66.8 to 69.5) | 49.4 (47.0 to 51.8) | – |
| | | Women | 66.5 (66.2 to 66.8) | 75.5 (75.3 to 75.7) | 75.1 (74.7 to 75.5) | 67.7 (65.5 to 69.8) | 73.6 (72.9 to 74.2) | 70.3 (69.7 to 70.8) | 78.0 (77.7 to 78.2) | 65.9 (63.9 to 67.9) | 48.9 (47.0 to 50.8) | – |
| Change | Sex | Men | 18.7 (18.5 to 19.2) | 18.6 (18.4 to 18.9) | 9.7 (9.1 to 10.3) | 13.9 (11.4 to 16.4) | 11.4 (10.7 to 12.2) | 0.2 (–0.7 to 1.0) | –7.5 (–7.8 to –7.2) | 1.7 (0.3 to 3.0) | –1.4 (–4.2 to 1.4) | – |
| | | Women | 21.3 (21.0 to 21.7) | 21.9 (21.6 to 22.1) | 13.1 (12.6 to 13.6) | 15.1 (12.2 to 17.9) | 12.2 (11.3 to 13.0) | 2.9 (2.4 to 3.5) | –6.8 (–7.0 to –6.5) | 2.9 (1.0 to 4.9) | –2.4 (–4.6 to –0.1) | – |

EQ VAS is the visual analogue scale component of the EQ-5D questionnaire.
BOA, Better management of patients with Osteoarthritis; NQRs, National Quality Registers.

## DISCUSSION
### Main findings
This study documented self-reported health status in different patient groups, and for specific diagnoses in the NQRs as well as in the general population. Higher proportions of problems and lower self-assessed EQ VAS scores (worse overall HRQoL) were reported among patients with diagnoses with more symptoms experienced captured by the EQ-5D-3L dimensions. In general, most problems were reported in the pain/discomfort dimension.

### Comparisons with other studies
The proportions of any problems reported in the EQ-5D-3L dimensions, and distribution between *some* and *severe* levels of problem, seem to vary depending on the differences in the symptoms of the diseases as well as the way in which they are experienced by the patients.[30–34] Specifically, patients in the intervention-based registers, such as spine, hip and knee, showed the highest proportions of problems in the EQ-5D-3L dimensions. This was particularly the case regarding the pain/discomfort dimension, which reflects the pain symptoms associated with the diseases.[31–34] Pain/discomfort was similarly the dimension with the highest proportion of problems reported, also in the general population. Substantially, less reported problems were noted after surgical interventions in patients from intervention-based registers. The high proportions of problems within the mobility and usual activities dimensions in COPD and patients with pulmonary fibrosis might be due to limitations in mobility and other functions and, in some cases, the need to use breathing equipment. Other studies of patients with respiratory diseases also reported comparably high proportions of problems within the mobility, usual activities and pain/discomfort dimensions.[30 35]

The EQ-5D-3L dimensions for which higher proportions of problems were reported seem to be partly determined by the type of symptoms/manifestations of the disease in patients, as indicated above. For instance, when comparing spine and psoriasis patients, the proportion and severity of problems in both the pain/discomfort and mobility dimensions were much lower for the patients with psoriasis. This provided evidence for the ability of the EQ-5D-3L to discriminate among conditions that affect specific aspects of health differently, although some dimensions more relevant to some diseases may not be covered. Even in comparison with patients with heart failure, who were on average much older than spine patients, higher proportions of problems with pain/discomfort and mobility were reported in spine patients. The low prevalence of severe problems ('confined to bed') with mobility might indicate the importance of applying the five-level EQ-5D-5L among these patient groups.[36]

Regarding the anxiety/depression dimension, in most of the registers fewer than half of the patients reported problems, with the exception of spine, cruciate ligament

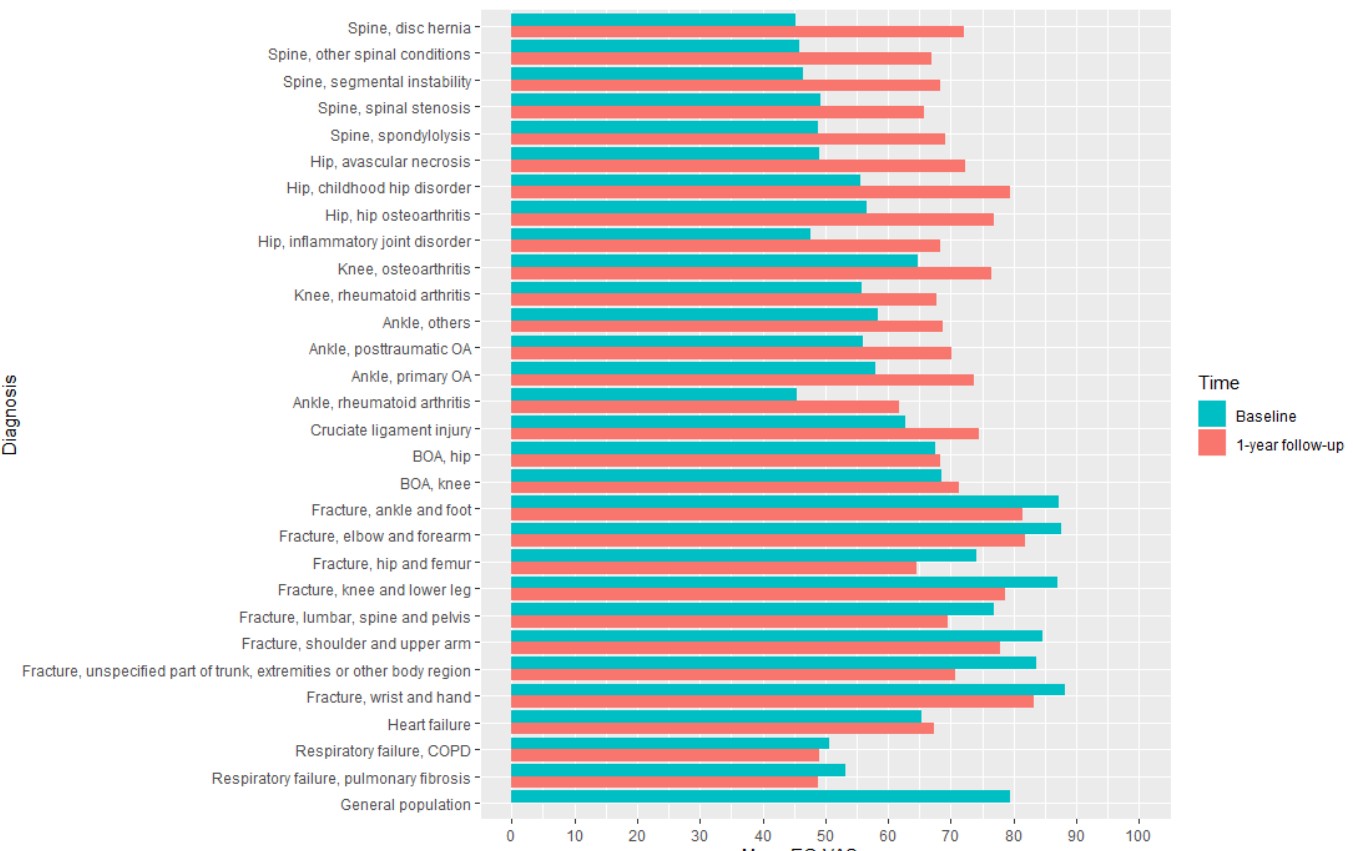

**Figure 2** Mean EQ VAS score by diagnosis at baseline and 1-year follow-up in the nine National Quality Registers (NQRs) and the general population. BOA, Better management of patients with Osteoarthritis; COPD, chronic obstructive pulmonary disease; OA, osteoarthritis. EQ VAS is the visual anaogue scale component of the EQ-5D questionnaire.

injury, respiratory failure and psoriasis patients at baseline, and respiratory failure patients at 1-year follow-up. Compared to the general population, the proportion of problems with anxiety/depression was higher in most of the registers at baseline. NQRs in which patients reported higher proportions of problems mainly on the pain/discomfort and mobility dimensions also had higher proportions of problems with anxiety/depression. This may be related to the pain, reduced walking ability and impaired ability to perform everyday activities they reported, which might affect the mood dimension. The interrelated nature of the EQ-5D dimensions has been reported for the EQ-5D-5L assessing causal and effect relationships, where empirical evidence showed mobility to be a causal indicator.[37] The highest proportions of problems in the mood dimension were reported among respiratory and spine patients at baseline and 1-year follow-up, and in rheumatology patients at 1-year follow-up. This could be related to the very restrictive nature of the diseases.[31 35 38] Other studies have also shown that more than half of patients with severe COPD, as well as those undergoing spinal surgery, report problems on the anxiety/depression dimension.[39 40]

When comparing problems reported among patients in the different registers on the EQ-5D-3L dimensions to the general population, one must take into account that there could be comorbid illnesses among patients and morbidities in the general population. Inline with this, public health survey reports and studies showed that, among the general population sample in the present study, considerable levels of different health problems were reported including cardiovascular diseases, diabetes, mental health problems such as depression, reports of pain in the neck, shoulders, arms and back as well as allergic eye and nose problems.[41–43]

The longitudinal change in reported problems in the dimensions showed that, while improvement was observed in most of the NQRs, larger improvements were reported in intervention-based registers. These patients also reported the highest proportions of problems at baseline and were subsequently provided with surgical interventions. Among these were hip, ankle, knee and spine patients. This pattern was also observed in the specific diagnoses within the various patient groups and PCHC categories. PCHC has been employed in other studies to document change over time in various diseases, such as heart failure, prostate cancer, diabetes mellitus and overactive bladder.[44–48] The change for the fracture patients cannot be compared directly to changes for other patient groups due to the recall technique used to value health state before injury.

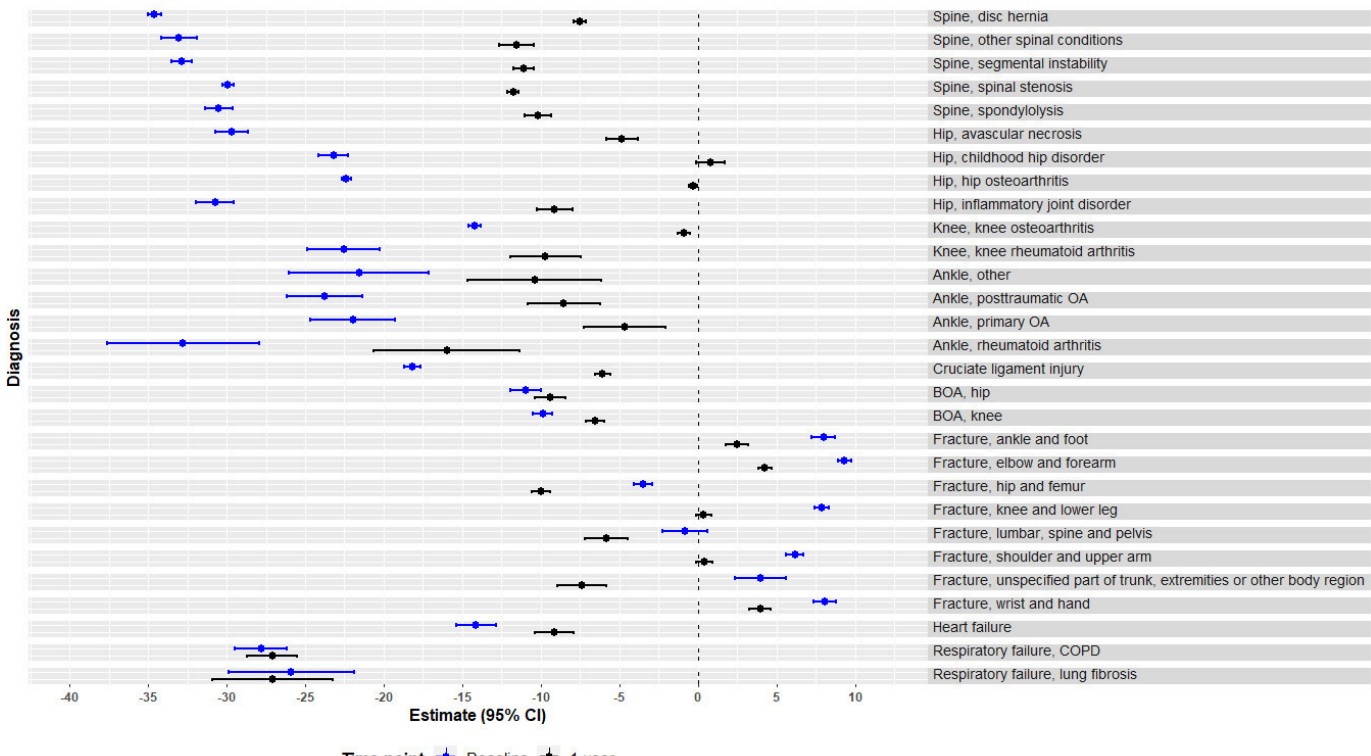

**Figure 3** Two-level random intercept model estimates of EQ VAS score by diagnosis adjusted for age and sex at baseline and 1 year in the nine National Quality Registers (NQRs) [reference group: general population]. BOA, Better management of patients with Osteoarthritis; COPD, chronic obstructive pulmonary disease; OA, osteoarthritis. EQ VAS is the visual anaogue scale component of the EQ-5D questionnaire.

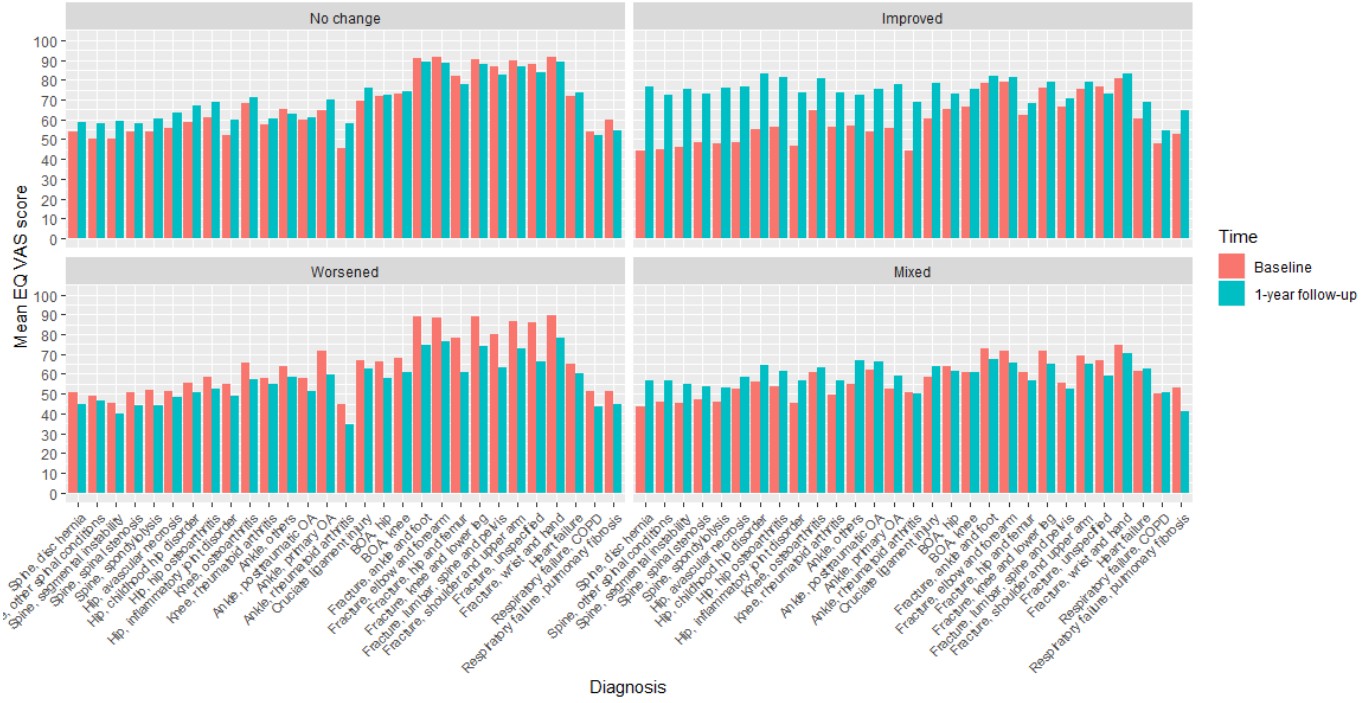

**Figure 4** Mean EQ VAS score by Paretian Classification of Health Change (PCHC) by diagnosis in the nine National Quality Registers (NQRs). BOA, Better management of patients with Osteoarthritis; COPD, chronic obstructive pulmonary disease; OA, osteoarthritis. EQ VAS is the visual anaogue scale component of the EQ-5D questionnaire.

The mean self-assessed EQ VAS scores showed variation across NQRs and diagnoses, with the lowest scores reported among patient groups with symptoms captured by the EQ-5D-3L dimensions. In almost all registers, the EQ VAS scores were lower, both at baseline and at 1-year follow-up, compared to findings in the general population data, for which the mean score was 79.5.[24] The EQ VAS scores from the registers demonstrated consistency with problems reported in the EQ-5D-3L dimensions. For example, spine and respiratory failure patients, who reported the highest proportions of problems at baseline, also had the lowest mean EQ VAS scores. In a study that assessed the performance of EQ-5D among patients with irritable bowel syndrome, EQ VAS score also was found to differ by level of severity.[49] The change in mean EQ VAS score also corresponded with changes in the problems reported in the descriptive system. This was shown by the largest increases in EQ VAS score among spine and hip patients. As the baseline level determines the room for improvement, the low baseline scores among these patient groups may have partly contributed to the large increments shown. The findings were in line with previous studies demonstrating the consistency of the EQ-5D descriptive system with the mean EQ VAS score.[4 50]

The consistent pattern of changes in the prevalence of problems reported in the EQ-5D-3L and the corresponding mean EQ VAS scores were also shown in the PCHC categories. This was particularly evident in an increase in the proportion of patients who were classified as *improved*, and a decrease in the proportion classified as *worsened*, in terms of EQ VAS scores. However, as also discussed in previous studies,[4 51] a pattern which might partly be attributed to the broader construct in EQ VAS than the EQ-5D-3L descriptive system was shown in the *no problem* category. In this category, mean EQ VAS scores increased over the 1-year follow-up, though they differed by diagnosis. The PCHC is potentially useful as a simple means of summarising changes using profile data without need for a value set. But, if there is a high proportion of patients classified as *mixed*, the method is less useful. PCHC does not inform on the magnitude of improvement or worsening in HRQoL.

The lower mean EQ VAS scores, consistently lower among women in most of the NQRs, were similar to findings reported in the literature among patients with different diseases as well as in the general population.[52–54] However, comparable findings on EQ VAS by sex were also reported.[55] Greater improvement was nevertheless found among women in many of the patient groups over the period up to the 1-year follow-up.

The variation in EQ VAS scores may be due to differences in the symptoms experienced in the different patient groups, as indicated by age-adjusted and sex-adjusted two-level random intercept model estimates. In most diagnoses, lower EQ VAS score than in the general population was shown at baseline, and these improved at 1-year follow-up. The mean EQ VAS score across diagnoses also showed a general pattern of more similar scores among diagnoses within the same NQR (patient group) than those across the different NQRs. One possible reason for this could be attributed to the similarity of diagnoses and symptoms in the same register.

## Strengths and limitations of the study

One of the strengths of the present study is the coverage of several diagnoses across different patient groups in the NQRs. The use of large data sets, providing information on the HRQoL of patients, together with data from the general population, made it possible to conduct relevant comparisons. Furthermore, the study assessed longitudinal change in HRQoL of patients from baseline to 1-year follow-up, making comparisons across the different NQRs. In addition, the study adds new data on patients with diagnoses that have previously been less studied. One limitation of the present study may be related to possible differences in the reported HRQoL arising due to different modes of data collection employed across NQRs. A second limitation relates to comorbidities among patients beside the main diagnoses and morbidities in the general population and how this may affect the comparison in the HRQoL across patient groups and with the general population data, not addressed fully in the present study. Another limitation relates to the inclusion of patients having both data at baseline and at 1-year follow-up, while excluding non-respondents, those with only baseline or 1-year follow-up data, and patients with incomplete entries. This may introduce bias. However, separate analyses of baseline data (regardless of 1-year follow-up) and 1-year follow-up data (regardless of baseline status) showed findings consistent with those included in the main analysis. Furthermore, in a study among patients who underwent total hip replacement, 1-year and 6-year follow-up data showed generally comparable prevalence of problems in EQ-5D-3L dimensions and similar mean EQ VAS scores.[56] Analyses of non-responders at the 1-year follow-up showed statistically significant but small differences in sex and age, as compared with respondents with follow-up data in many registers.

## Implications of the findings

The present study contributes by documenting HRQoL among several patient groups, as compared with the general population. It adds to the literature in the area where studies covering various diagnoses have not been widely conducted. By using routinely collected comprehensive data from a number of NQRs in Sweden, the study also adds to previous findings on the consistency between problems reported on the EQ-5D-3L dimensions and the EQ VAS. This is a useful addition to the literature, showing the importance of using the EQ VAS to describe overall health from the patient perspective, using relatively concise data. This can be valuable in monitoring the health status and progress of different subgroups (age, sex, clinical stages) of patients. The present study demonstrated an important feature of the EQ-5D-3L dimensions and the EQ VAS, showing differences across diagnoses.

## CONCLUSIONS

The present study contributes to the literature by documenting self-reported health status for several different patient groups and diagnoses using data collected through the EQ-5D-3L questionnaire, and comparing these with data on the general population. The consistency between problems reported on the EQ-5D-3L dimension and the EQ VAS score demonstrated component corresponding patterns at baseline and at 1 year follow-up.

This study shows the important role of PROs in routine clinical care, to assess and follow-up health status and progress within different groups of patients. Together with clinical data, this could provide crucial information for understanding and improving the patients' health status.

**Author affiliations**

[1]Health Outcomes and Economic Evaluation Research Group, Stockholm Centre for Healthcare Ethics, Department of Learning, Informatics, Management and Ethics, Karolinska Institutet, Stockholm, Sweden

[2]Department of Orthopaedics, Institute of Clinical Sciences, Sahlgrenska Academy, University of Gothenburg, Gothenburg, Sweden

[3]Swedish Hip Arthroplasty Register, Gothenburg, Sweden

[4]Health Economics, The University of Melbourne School of Population and Global Health, Melbourne, Victoria, Australia

[5]Office of Health Economics, London, UK

[6]City University of London, London, UK

[7]Equity and Health Policy Research Group, Department of Global Public Health, Karolinska Institutet, Stockholm, Sweden

[8]Health Care Services, Region Stockholm, Stockholm, Sweden

**Collaborators** The Swedish Quality Register (SWEQR) Study Group: Allan Abbott, Associate Professor, Linköping University, Linköping, Sweden. Magnus Ekström, Associate Professor, Lund University, Lund, Sweden. Magnus Forssblad, Associate Professor, Karolinska Institutet, Stockholm, Sweden. Peter Fritzell, Associate Professor, Futurum Academy for Health and Care, Jönköping, Sweden/ RKC Centre for spine surgery, Stockholm, Sweden. Åsa Jonsson, Ryhov County Hospital, Jönköping, Sweden. Mikael Landén, Professor, University of Gothenburg, Gothenburg, Sweden. Michael Möller, Associate Professor, University of Gothenburg, Gothenburg, Sweden. Malin Regardt, Reg OT, PhD, Swedish Rheumatology Quality Register (SRQ), Karolinska University Hospital, Stockholm; Karolinska Institutet, Stockholm, Sweden. Björn Rosengren, Professor, Lund University, Lund, Sweden. Marcus Schmitt-Egenolf, Professor, Umeå University, Umeå, Sweden. Johanna Vinblad, Project leader, Centre of Registers, Gothenburg, Sweden. Annette W-Dahl, Associate Professor, Lund University, Lund, Sweden.

**Contributors** KB and OR, conceived the study. FST, OR, KB, ND, DP, EN and members of the SWEQR Study Group (AA, ME, MF, PF, ÅJ, ML, MM, MR, BR, MS-E, JV, AW-D) designed the study. FST performed the data analysis and preliminary interpretation of the data. KB, OR, ND, DP and EN supervised data analysis and interpretation of findings. FST drafted the manuscript. KB, OR, ND, DP, EN and members of the SWEQR Study Group (AA, ME, MF, PF, ÅJ, ML, MM, MR, BR, MS-E, JV, AW-D) revised the draft manuscript for important intellectual content. The corresponding author attests that all the authors in the list fulfil the authorship criteria and no other who fulfil the criteria were left out. All authors approved the submission of the manuscript. All the authors agree to be responsible for all aspects of the manuscript. FST takes responsibility for the accuracy of the data analysis and the overall content of the manuscript.

**Funding** This research project is supported by a grant from The EuroQol Research Foundation (EQ Project number 2016480) and Region Stockholm (former Stockholm County Council) (number 4-3464/2018), Stockholm, Sweden. The funding bodies did not influence the design, conduct and reporting of the study.

**Competing interests** KB, ND and DP are members of the EuroQol Group. KB reports grants from EuroQol Research Foundation, grants from Region Stockholm, during the conduct of the study; personal fees from Region Stockholm, outside the submitted work. ML reports personal fees from Lundbeck Pharmaceuticals, outside the submitted work. OR reports institutional compensation for educational consultancy from Link Sweden; institutional compensation for research consultancy from Pfizer, outside the submitted work. AA, ME, MF, PF, ÅJ, MM, EN, MR, BR, MS-E, FST, AW-D and JV declare no competing interests.

**Patient consent for publication** Not applicable.

**Ethics approval** Ethical approval (number 1185-18/2019-00812 and number 2020-04369) was obtained from the Regional Ethics Review Board in Gothenburg, Sweden, for the data from the National Quality Registers and from the Swedish Ethical Review Authority (number 2020-03090) for the general population data.

**Provenance and peer review** Not commissioned; externally peer reviewed.

**Data availability statement** No data are available. Data sharing is not possible according to Swedish law.

**ORCID iD**
Fitsum Sebsibe Teni http://orcid.org/0000-0002-6182-499X

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
