## [Reviewer comments · BMJ Open]

ARTICLE DETAILS

TITLE (PROVISIONAL)	A longitudinal study of patients' health-related quality of life using EQ-5D-3L in 11 Swedish National Quality Registers
AUTHORS	Teni, Fitsum; Rolfson, O; Devlin, Nancy; Parkin, David; Naclér, Emma; Burström, Kristina; the Swedish Quality Register (SWEQR) Study Group, na

VERSION 1 – REVIEW

REVIEWER	Yang, Zhihao Guizhou Medical University
REVIEW RETURNED	28-Feb-2021

GENERAL COMMENTS	Thank you for inviting me to review this manuscript. I think it is an amazing study considering the amount of the data included in the study, good job! Of course, a major difficulty comes with the large data is the number and size of the tables. It takes a lot of time to go through all the tables! Possibly due to the large number of tables, the analysis only limited to descriptive analysis, which is unusual for a scientific study. I would suggest the authors to consider dropping some information and focus on the main research question, for example, I do not think reporting the number of unique health states is important for readers; discussing the relationship between VAS and health state, note the manuscript spent a paragraph to discuss about this issue, I think this is more of a methodological concern that will not interest many readers etc. A solution might be to break the current manuscript into two, one focus on the change of health, and one focus on EQ-5D instrument.
--

REVIEWER	Thompson, Alexander 3. Manchester Centre for Health Economics, The University of Manchester, Manchester, UK, M13 9PL.
REVIEW RETURNED	29-Apr-2021

GENERAL COMMENTS	This is a comprehensive paper but I fear it is trying to do too many things at once, which in turn detracts from the individual components within the work. It is not entirely clear to me whether the paper is merely seeking to be descriptive by documenting the HRQoL of the various conditions, within the various registers, using the EQ-5D-3L (there are many tables doing that) and comparing that with a general population sample. Alternatively, the paper also seems to have a methodological slant by exploring whether the EQ-VAS gives comparable scores and changes to the EQ-5D-3L. Comparisons between the EQ-5D and the VAS has been a strangely under-
---

	researched area so there is definitely merit in exploring the relationship more but I wonder whether BMJ Open is the place for it? Running through both the descriptive and the methodological aspects are comparisons made with a general population sample, although that sample does not seem to be well described, apart from some fairly simple descriptive characteristics reported in Table 1. In total there are 6 tables and 13 supplementary tables so it seems somewhat remiss not to provide some further information on that particular cohort rather than linking to another paper. Specifically, I would be interested in the reported morbidities. The analysis seems appropriate and the key findings reasonable but if the paper is to do with comparisons between the disease-specific registers and a general population sample I would have liked to have seen better adjustment for sex and age differences and some commentary around the morbidities experienced between the samples (presumably patients on the disease register do not just have one health problem and presumably those in the general population have problems too). Alternatively, if the paper is focused on the EQ-5D versus VAS I would have liked to have seen more explicit analysis there. For example, large differences in VAS scores for the same health state, for certain disease registers, what does that mean? Why does it differ a lot in some registers and not in others? I did not really follow the regression element and believe that could have been described better in the body or SA. Please define how the sample was pooled and give the full model, sample size and coefficients etc in the SA. Moreover, table after table can be overwhelming for the visual learners out there, so I would suggest trying to present the data in different ways and also selecting the most important elements. Overall this is a high quality and thorough piece of analysis but I would suggest narrowing the scope and focusing on either the descriptive or the methodological aspects.
--	---

VERSION 1 – AUTHOR RESPONSE

Reviewer 1

Thank you for inviting me to review this manuscript. I think it is an amazing study considering the amount of the data included in the study, good job! Of course, a major difficulty comes with the large data is the number and size of the tables. It takes a lot of time to go through all the tables! Possibly due to the large number of tables, the analysis only limited to descriptive analysis, which is unusual for a scientific study. I would suggest the authors to consider dropping some information and focus on the main research question, for example, I do not think reporting the number of unique health states is important for readers; discussing the relationship between VAS and health state, note the manuscript spent a paragraph to discuss about this issue, I think this is more of a methodological concern that will not interest many readers etc. A solution might be to break the current manuscript into two, one focus on the change of health, and one focus on EQ-5D instrument.

- We have now taken out a number of tables from the main text and the supplementary materials to reduce the size of the manuscript. These are Tables 6 and 7 in the main manuscript and Tables S3, S7, S8, S9, S10 and S11 in the supplementary material.
- We have focused the manuscript toward reporting of HRQoL of different patient groups in Sweden by taking out aspects focusing on the methodological issues.
- We have also taken out texts and part of the table on unique health states and the methodological aspect of the relationship between VAS and health state. These specific changes are indicated below.
 - In the Introduction section of the original submission, texts focused on the methodological aspect of the relationship between EQ-5D-3L and EQ VAS score are now taken out. Specifically, on page 5 lines 16 to 14 are now taken out. In the same page the text between lines 33 and 40 are now taken out.
 - In the Introduction section of the original submission, the objective focused on comparing EQ VAS score across selected health states is now taken out (page 7, line 2 to 5).
 - In the Methods section, data analysis subsection of the original submission (page 9, lines 29 to 32), a sentence on analysis of the selected nine health states is taken out.
 - In the Results section of the original submission, in paragraph 1, the information on the number of health states is taken out (page 11, lines 9 to 12).
 - In the Results section of the original submission, the Pareitian Classification of Health Change subsection, the sentence regarding the number of health states is taken out (page 12, lines 33 to 35).
 - In the Results section of the original submission, the Mean self-assessed EQ VAS score subsection, a paragraph on results of EQ VAS score for the nine selected health states are taken out (page 14, lines 16 to 35).
 - In the Discussion section, the paragraph just before Strengths and limitations subsection in the original submission (page 18, lines 3 to 28), focused on the methodological aspect of the EQ-5D through the nine selected health states is, now taken out.
 - In the Discussion section, Implications of the findings subsection of the original submission, the last two sentences focusing on the methodological aspect of EQ VAS are now taken out (page 19, lines 26 to 33).
 - In the Conclusions section, the second paragraph in the original submission is now taken out (page 20, lines 15 to 24).

Reviewer 2

This is a comprehensive paper but I fear it is trying to do too many things at once, which in turn detracts from the individual components within the work. It is not entirely clear to me whether the paper is merely seeking to be descriptive by documenting the HRQoL of the various conditions, within the various registers, using the EQ-5D-3L (there are many tables doing that) and comparing that with a general population sample. Alternatively, the paper also seems to have a methodological slant by exploring whether the EQ-VAS gives comparable scores and changes to the EQ-5D-3L. Comparisons between the EQ-5D and the VAS has been a strangely under-researched area so there is definitely merit in exploring the relationship more but I wonder whether BMJ Open is the place for it?

□ We have now narrowed the manuscript by focusing it on describing the HRQoL of different patient groups and comparing it with the general population. The specific texts and tables taken out are detailed above.

Running through both the descriptive and the methodological aspects are comparisons made with a general population sample, although that sample does not seem to be well described, apart from some fairly simple descriptive characteristics reported in Table 1. In total there are 6 tables and 13 supplementary tables so it seems somewhat remiss not to provide some further information on that particular cohort rather than linking to another paper. Specifically, I would be interested in the reported morbidities.

□ We have now added a paragraph in the Methods section of the manuscript describing the general population data. This is available in the first paragraph of page of 8 of the revised manuscript.

The analysis seems appropriate and the key findings reasonable but if the paper is to do with comparisons between the disease-specific registers and a general population sample I would have liked to have seen better adjustment for sex and age differences and some commentary around the morbidities experienced between the samples (presumably patients on the disease register do not just have one health problem and presumably those in the general population have problems too).

□ In the analysis we have provided information on changes in EQ VAS score over the follow-up using analysis of covariance which made comparison by sex while adjusting for age. We have now brought this table into the main manuscript text.

□ In addition, the regression analysis (two-level random intercept model) looking into the influence of diagnosis on EQ VAS score was performed by adjusting for sex and age both at baseline and at 1-year follow-up.

□ We have now added figures, for the proportions of problems reported on EQ-5D-3L, stratified by sex and by age groups for all the patient groups and the general population. This is performed for both baseline as well as 1-year follow-up data. These results are now presented in the supplementary material Figures S1 and S2.

□ In relation to morbidities reported in the general population and the patient groups, we have now added in the discussion the nature of self-reported morbidities in the general population surveys citing the public health reports in the discussion section. This is now added on page 15, paragraph 2. In relation to morbidities in the different patient groups, as establishing of a complete comorbidity profile of the different patient groups would require linking to other registers on morbidities of patients from different types and levels of care, we have indicated this as one of the limitations in the comparison of the patient groups and the general population. This is presented on page 17 in the Strengths and limitations subsection.

Alternatively, if the paper is focused on the EQ-5D versus VAS I would have liked to have seen more explicit analysis there. For example, large differences in VAS scores for the same health state, for certain disease registers, what does that mean? Why does it differ a lot in some registers and not in others?

□ As indicated above we have now focused the manuscript on the reporting on the HRQoL of the different patient groups. The specific texts and tables taken out and those modified are indicated above.

I did not really follow the regression element and believe that could have been described better in the body or SA. Please define how the sample was pooled and give the full model, sample size and coefficients etc in the SA.

□ We have now clarified the procedure followed in the regression analysis. We have added this information in the analysis subsection of the methods section on page 8 of the revised manuscript.

□ We have also presented a table version of the analysis output in the supplementary material as Table S5.

Moreover, table after table can be overwhelming for the visual learners out there, so I would suggest trying to present the data in different ways and also selecting the most important elements.

□ We have now converted Tables 4 and 5 into Figures (now Figure 2 and Figure 4).

□ We have also used figures to present the added analysis of problems reported in the EQ-5D-3L dimensions stratified by age groups among men and women in the supplementary materials file (Figures S1 and S2).

Overall this is a high quality and thorough piece of analysis but I would suggest narrowing the scope and focusing on either the descriptive or the methodological aspects.

□ Thank you, we have accordingly focused the manuscript on reporting HRQoL of different patient groups in Sweden.

VERSION 2 – REVIEW

REVIEWER	Thompson, Alexander 3. Manchester Centre for Health Economics, The University of Manchester, Manchester, UK, M13 9PL.
REVIEW RETURNED	15-Oct-2021
GENERAL COMMENTS	The authors have substantially improved the paper from the first draft and focused the scope of the paper. I am satisfied that all suggestions have been appropriately incorporated into the second iteration.